# Setting the photoelectron clock through molecular alignment

Andrea Trabattoni[1,2], Joss Wiese [1,3], Umberto De Giovannini [4], Jean-François Olivieri[1], Terry Mullins[1], Jolijn Onvlee [1], Sang-Kil Son [1,2], Biagio Frusteri [5], Angel Rubio [4,6,7✉], Sebastian Trippel[1,2] & Jochen Küpper [1,2,3,7✉]

The interaction of strong laser fields with matter intrinsically provides a powerful tool for imaging transient dynamics with an extremely high spatiotemporal resolution. Here, we study strong-field ionisation of laser-aligned molecules, and show a full real-time picture of the photoelectron dynamics in the combined action of the laser field and the molecular inter-action. We demonstrate that the molecule has a dramatic impact on the overall strong-field dynamics: it sets the clock for the emission of electrons with a given rescattering kinetic energy. This result represents a benchmark for the seminal statements of molecular-frame strong-field physics and has strong impact on the interpretation of self-diffraction experiments. Furthermore, the resulting encoding of the time-energy relation in molecular-frame photoelectron momentum distributions shows the way of probing the molecular potential in real-time, and accessing a deeper understanding of electron transport during strong-field interactions.

[1] Center for Free-Electron Laser Science, Deutsches Elektronen-Synchrotron DESY, Notkestraße 85, 22607 Hamburg, Germany. [2] The Hamburg Center for Ultrafast Imaging, Universität Hamburg, Luruper Chaussee 149, 22761 Hamburg, Germany. [3] Department of Chemistry, Universität Hamburg Martin-Luther-King-Platz 6, 20146 Hamburg, Germany. [4] Max Planck Institute for the Structure and Dynamics of Matter and Center for Free-Electron Laser Science, 22761 Hamburg, Germany. [5] Dipartimento di Fisica e Chimica, Universitá degli Studi di Palermo, Via Archirafi 36, 90123 Palermo, Italy. [6] Center for Computational Quantum Physics (CCQ), The Flatiron Institute, 162 Fifth Avenue, New York, NY 10010, USA. [7] Department of Physics, Universität Hamburg, Luruper Chaussee 149, 22761 Hamburg, Germany. ✉email: angel.rubio@mpsd.mpg.de; jochen.kuepper@cfel.de

In the prototypical strong-field interaction, an intense driving field extracts a valence electron from the target through tunnel ionisation, accelerates the free electron in vacuum and eventually drives it back to the parent ion, predominantly resulting in rescattering or radiative recombination[1,2]. The radiative recombination results in the emission of high-energy photons by high-harmonic generation[1], and this is a powerful tool to investigate the electronic structure with attosecond temporal resolution[3–5]. Alternatively, the rescattered portion of this electron wavepacket is exploited in laser-induced electron diffraction (LIED)[6] experiments as a coherent diffraction pattern of the molecular target, potentially providing time-dependent images of the molecule at sub-femtosecond and few-picometer resolution. Recently, corresponding experimental results for the structure or dynamics of small or highly symmetric molecules were obtained[7–12]. At the same time, the initial conditions of the strong-field interaction have attracted much attention for capturing the intrinsic nature of strong-field physics.

While pioneering attosecond experiments and molecular-frame measurements revealed non-trivial spatiotemporal features in electron tunnelling[13,14], these initial conditions are still generally considered a weak perturbation in strong-field physics. All the results obtained in LIED experiments, for example, are interpreted in the framework of the strong-field approximation, where the electron is considered to be born in the continuum with a negligible initial momentum, and to propagate as a plane wave[15]. Furthermore, the post-ionisation dynamics before rescattering are assumed to be fully driven by the laser field, by neglecting, for example, the Coulomb interaction with the ionised molecule.

Common strategies to analyse photoelectron-momentum distributions rely on the quantitative rescattering theory (QRS)[15], where angular dependence in the final photoelectron wavepacket is introduced solely through rescattering. Within this approach, diffraction patterns were analysed utilising the angular[7,8] or radial[16] photoelectron distribution. However, the relevance of the ionised molecular orbital in the rescattered photoelectrons is still under discussion[17]. So far, this was included by an overall weighting factor in the rescattering probability[18,19], or as a spatial phase or an angular feature in the rescattering electron wavepacket[14,20]. Recently, the influence of molecular alignment on molecular structure retrieval was discussed[16,21]. However, general predictions are still extremely challenging with new models appearing[22,23].

Here, we experimentally and computationally study molecular-frame photoelectron spectroscopy from strongly aligned molecules in order to investigate the relation between the molecular frame and the strong-field-induced ultrafast electron dynamics. We demonstrate that and how the molecular frame governs the rescattering time for the photoelectron and, consequently, its final kinetic energy.

## Results

**Experimental approach**. Figure 1 depicts the experiment. An ensemble of carbonyl sulfide (OCS) molecules all in the rovibronic ground state[24] was adiabatically aligned in the laboratory frame, with $\cos^2\theta_{2D} = 0.9$, by using a linearly polarised, 500 ps laser pulse, centred at 800 nm[25,26], with a peak intensity $I = 3 \times 10^{11}$ W cm$^{-2}$. The molecules were aligned in two different configurations, shown in Fig. 1, with the molecular axis along the $Y$ and $Z$ axes, named parallel and perpendicular alignment, respectively. A second laser pulse, centred at 1300 nm, with a duration of 65 fs, and a peak intensity $I = 8 \times 10^{13}$ W cm$^{-2}$, was used to singly ionise the OCS molecules. For this intensity, the ponderomotive energy of the laser field is $U_p \approx 13$ eV and the ionisation occurred in the tunnelling regime. The electric field of

the ionising laser pulse, $\mathbf{E}_L$ in Fig. 1, was linearly polarised along the $Y$ axis (ellipticity $\epsilon = I_Z/I_Y < 0.005$). The produced molecular-frame angle-resolved photoelectron spectra (MF-ARPES) were recorded in a velocity map-imaging (VMI) spectrometer[27] with its detector parallel to the $XY$ plane. It is important to note that the de Broglie wavelength of rescattering electrons in the experiment was larger than 200 pm. In this regime no diffraction feature is expected to appear in the photoelectron distributions[16].

**Photoelectron-momentum distributions**. Figure 1 shows the MF-ARPES for parallel (left) and perpendicular (right) alignment. The two distributions show several differences. The spectrum for parallel alignment has a larger width at small transverse momenta, $p_X < 0.5$ a.u. (atomic units), while the spectrum for perpendicular alignment shows a number of angular features for transverse momenta $p_X$ between 0.5 and 1 a.u. These angular structures, which are much weaker in the spectrum for parallel alignment, could be identified as forward-rescattering features[28]. Focussing the attention on large longitudinal momenta $p_Y$, the counts for parallel alignment drop around 2.5 a.u. In the case of perpendicular alignment, however, the spectrum extends to larger momenta, showing an appreciable amount of counts at $p_Y = 3$ a.u. Following the strong-field approximation, the hard cut-off of photoelectron momentum is expected to only depend on the properties of the laser field[29]. Experimentally, this quantity is hard to measure. Thus, the turning-point of the signal drop, i.e., the minimum of the first derivative, at large longitudinal momenta is used instead. In the following, we use the term cut-off in the latter sense. Surprisingly, in the current study we found a clear dependence of the cut-off on the molecular frame.

Figure 2a shows a close comparison of the two experimental distributions for the complete range of $p_X$ and $p_Y$, between 0 and 4 a.u. Here, the spectra were split along the $Y$ axis and the spectrum from parallel alignment is shown on the left and the one from perpendicular alignment on the right. Now, the differences at small momenta as well as at the cut-off are even more evident. To perform a quantitative analysis of the cutoffs, the momentum distributions were angularly integrated within a cone of ±20° with respect to the longitudinal axis ($Y$) and converted to an energy scale. In Fig. 2b, the resulting photoelectron spectra are shown for parallel (blue) and perpendicular (red) alignment, with energies in units of $U_p$. The perpendicular/parallel ratio of the two area-normalised spectra (green) shows a predominance of photoelectrons for perpendicular alignment in the energy range between 2 and 10 $U_p$, where the distribution is dominated by rescattered electrons[30]. Furthermore, the ratio increases with energy, reaching the maximum around the cut-off. To evaluate the cutoffs, the first derivative of the energy distributions are shown in Fig. 2c and their minima were used to find the edges of the distributions, which allowed us to analyse the cut-off region. The first minimum represents the drop of direct electrons[30] and it was around 2 $U_p$ for both alignment cases. This excluded any significant alignment-dependent direct-electron-cut-off enhancement[31]. Surprisingly, the second minimum behaves differently for the two alignments. While it is located around 10 $U_p$ for perpendicular alignment, as expected from the well-established above-threshold ionisation theory[29], the cut-off is shifted down to a value around 8.5 $U_p$ for parallel alignment.

**Quantum-mechanical model of the electron dynamics**. To unravel the experimental observations, state-of-the-art calculations were performed using both, time-dependent density-functional theory (TDDFT)[32] and a semi-classical molecular trajectory simulation set-up. Using TDDFT, the MF-ARPES probability was calculated by simulating the complete dynamics

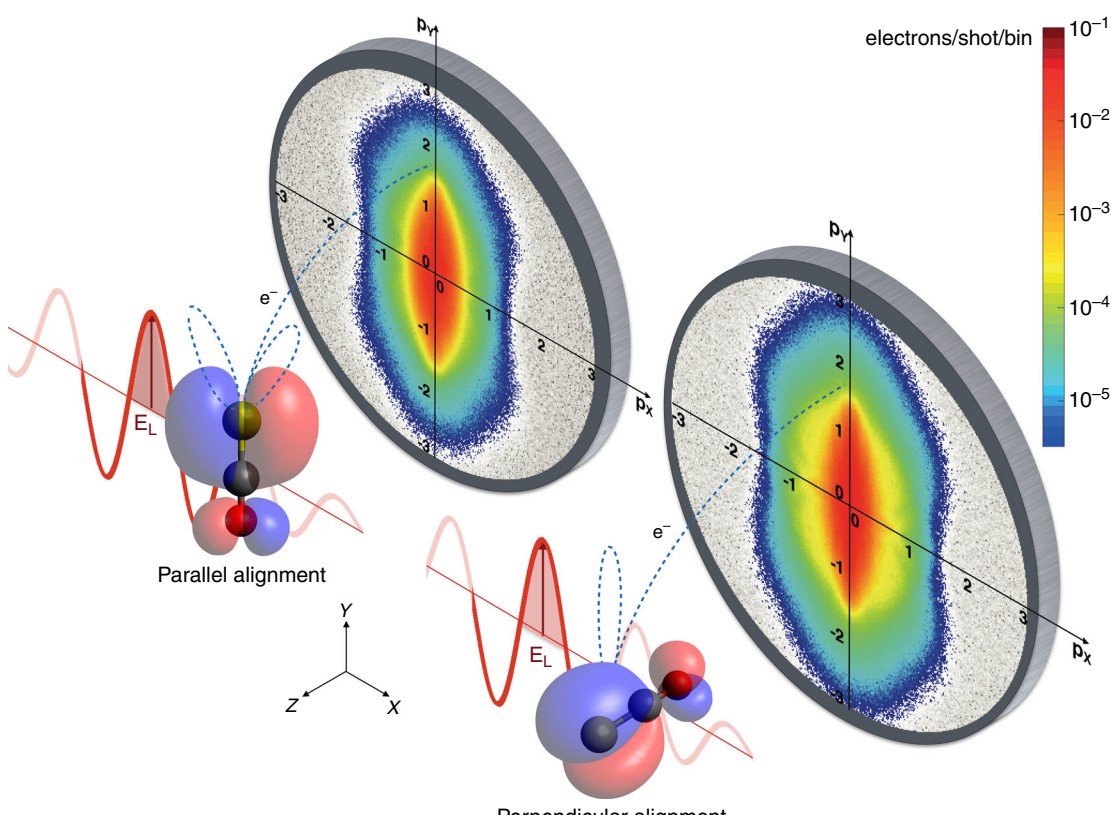

**Fig. 1 Sketch of the experimental arrangement.** OCS molecules (O in red, C in black, S in yellow) were aligned in the laboratory frame, parallel and perpendicular to the $Y$ axis. The ionising laser electric field ($\mathbf{E}_L$) was linearly polarised along the $Y$ axis and the detection was in the $XY$ plane. The molecular-frame angle-resolved photoelectron spectra were projected onto a 2D detector in a velocity map-imaging spectrometer. The alignment-dependent photoelectron trajectories are pictorially shown (blue dashed lines), as well as the corresponding shape of the ionising orbital (blue and red lobes). The spectra are displayed on a logarithmic intensity scale in units of electrons/shot/bin.

of the many-body ionisation process in real-time and real-space with the tSURFF method[33,34], see Supplementary Note 1 for details. With this technique the spectrum was obtained by computing the entire time-dependent electron dynamics, including many-body electron interactions, and collecting the flux of electrons through a closed surface surrounding the molecule. Figure 2d–f report the same analysis of the numerical results as performed for the experimental data in Fig. 2a–c. The simulations capture the principal features of experimental data very well. In particular, Fig. 2f shows that the calculations reproduce the experimental cut-off positions for parallel and perpendicular alignment as well as the corresponding shift between them very well. This result is strongly affected by the electron–electron interaction and the interplay between different orbitals. Indeed, it is evident from the calculation that the molecule is predominantly ionised from the highest-occupied molecular orbital (HOMO) for both alignments. In the case of parallel alignment, nevertheless, a small contribution of HOMO-1 to the yield of high-energy rescattered electrons is observed. When the electron–electron interaction is artificially turned off the HOMO-1 contribution becomes significant and in this scenario the reduced cut-off observed in the experiment is not reproduced, see Supplementary Fig. 4. Instead, in the case of fully interacting electrons the yield of the rescattered electrons ionised from HOMO-1 is suppressed, resulting in the really good agreement with the experiment.

**Semi-classical model of the electron dynamics.** Furthermore, semi-classical trajectory simulations based on the Ammosov–Delone–Krainov (ADK) tunnelling theory[35] in

conjunction with a simple man propagation (SM)[36,37] were conducted in order to track the molecular-frame electron dynamics during the strong-field interaction[38], see Supplementary Note 2 for details. Based on the TDDFT analysis of the different molecular orbitals contributing to the photoelectron dynamics, the ionisation was assumed to occur solely from HOMO. In the underlying model, the initial phase-space distribution of the electron wavepacket in the continuum at birth was described by the quasistatic ADK tunnelling theory, and the nodal structure of the HOMO was accounted for as an imprint onto this initial momentum distribution. Post-ionisation dynamics of the electron wavepacket were evaluated in the combined interaction with the laser electric field and the cation's Coulomb field modelled as a point charge. To evaluate the accuracy of this semi-classical description, the resulting MF-ARPES for parallel and perpendicular alignment was calculated and analysed (see Fig. 2g–i), and it shows a really good agreement, both, with the experimental data and the full TDDFT calculation, reproducing the main features and cutoffs observed in the experiment very well. In particular, as seen by the local maximum around 10 $U_p$ of the ratio of the two alignment cases (Fig. 2h), this semi-classical model captures the enhanced yield of high-energy rescattered electrons for perpendicular alignment with respect to parallel alignment. This result is corroborated by the enhanced cut-off around 10.7 $U_p$ for perpendicular alignment in Fig. 2i, although a smaller yield at this energy is present also for parallel alignment. In addition, a relevant minimum appears around 9.3 $U_p$ for both alignments. These features of ADK-SM, together with the pronounced yield along the centreline of Fig. 2g,

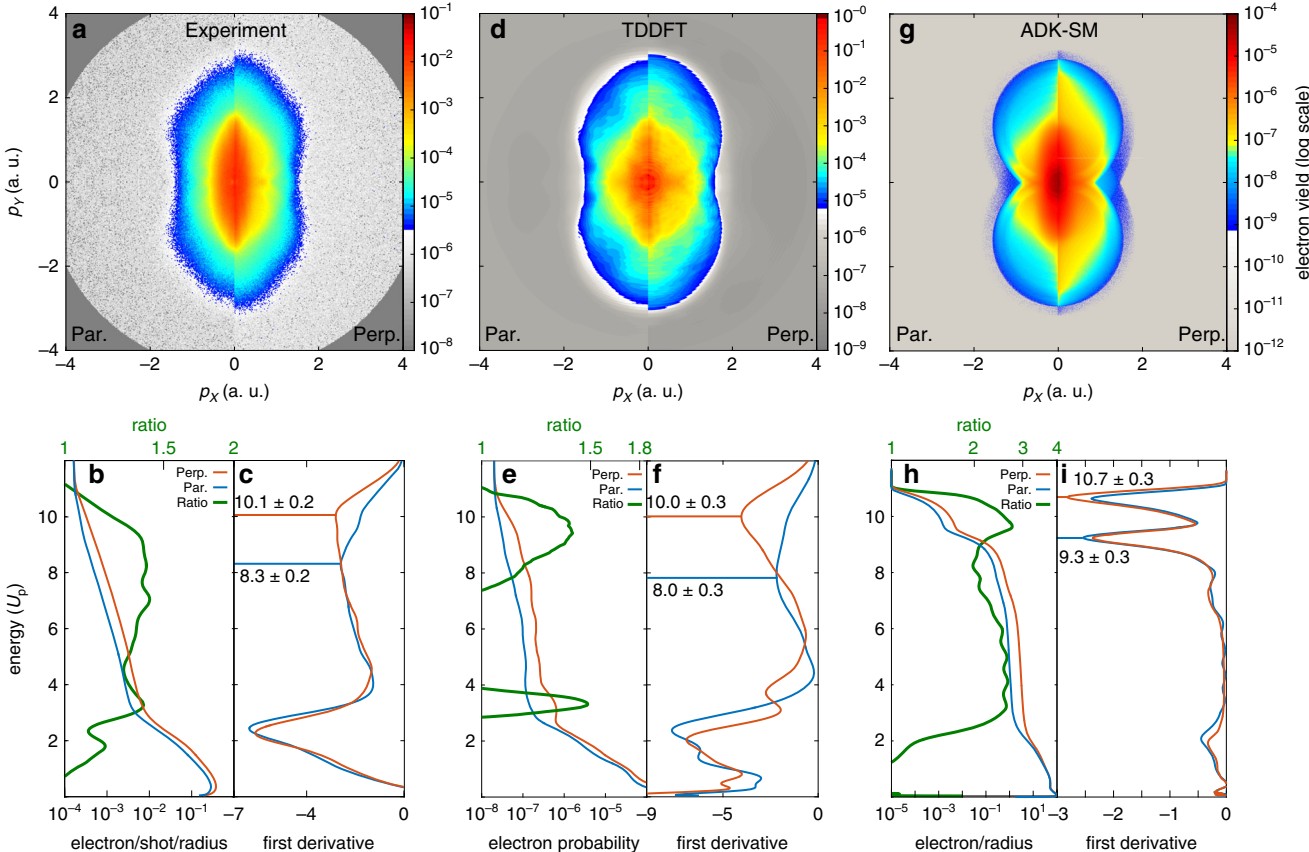

**Fig. 2 Molecular-frame angle-resolved photoelectron spectra of OCS.** These data were obtained **a–c** experimentally and computationally from **d–f** TDDFT and **g–i** ADK-SM calculations. **a**, **d**, **g** Split graphical representation as a comparison of the photoelectron distributions for parallel and perpendicular alignment for the experimental and computational results, respectively. **b**, **e**, **h** Corresponding projected energy distributions of photoelectrons along the $Y$ axis, angularly integrated within a cone of ±20°, as well as the ratio of the integral-normalised perpendicular and parallel distributions, on logarithmic scales. Energies are reported in units of the ponderomotive energy $U_p$. **c**, **f**, **i** First derivatives of the photoelectron-energy distributions to evaluate the high-energy cut-off for the two molecular-alignment cases. All TDDFT computational results were obtained by averaging over different laser-molecule orientations according to the experimental alignment distributions and by adding a constant to account for the experimental background level. The ADK-SM results refer to a single laser intensity and perfect alignment for both cases. See Supplementary Note 2 for details.

are known to be mainly due to Coulomb focussing[39], i.e., the dynamics of a continuum electron wavepacket being focussed along a perfectly linear laser polarisation axis. The relevance of this effect is discussed further below.

**Differential analysis of the momentum distributions**. To obtain a more comprehensive picture of the alignment-dependent photoelectron dynamics and, in particular, a glimpse at the initial electron wavepacket, we performed a differential analysis by subtracting the photoelectron distributions of the two alignment cases from each other. Figure 3a–c shows the relative normalised differences, parallel minus perpendicular, for the experiment, the TDDFT simulations, and the ADK-SM calculations, respectively. The agreement between experimental data and both models is excellent. Here, a strong depletion along the vertical axis and two transversely offset broad lines of positive yield appear as main features, with a really good agreement between experimental and both computational results. The depletion along the centreline is due to the nodal structure of the degenerate Π HOMO of OCS: it represents a forbidden direction of electron ejection[18,40]. Therefore, when the molecular axis was aligned along the polarisation axis of the strong field, the electron preferentially acquired an initial transverse momentum $p_{0X}$ that was much larger than in the case of perpendicular alignment, shown by the red vertical ridges in Fig. 3a–c.

## Discussion

The features observed in Fig. 3a–c show the crucial impact of the electronic structure on the initial conditions of the electron at birth. However, a quantitative evaluation of the initial conditions of the electron at tunnelling is challenging[41–43]. In general, they are defined by the tunnel-exit position as well as by the temporal phase and the momentum acquired during the ionisation with respect to the external field. Here, we demonstrate that the molecular potential, i.e., the combination of the electrostatic potential and the electronic structure of the molecule, has in fact not only a primary role in setting the initial conditions for electron emission, but that it also drives the whole photoelectron dynamics: it defines the cut-off of rescattered electrons and it shapes the time–energy relation for electron recollision. To investigate this, we exploited the ADK-SM calculations to analyse the final absolute momentum acquired by the electron after photoionisation as a function of the recollision phase. In the following discussion we will refer to revisit to describe passages of the electron nearby the cation on a relatively large spatial scale, where momentum transfer is relatively small and soft, whereas recollision, or rescattering, refers to the close approach of the electron to a nucleus, with an associated large-momentum transfer, e.g., in back scattering. While multiple revisits may occur during the interaction of the electron with its cation, a recollision event will drive the electron irreversibly away from the

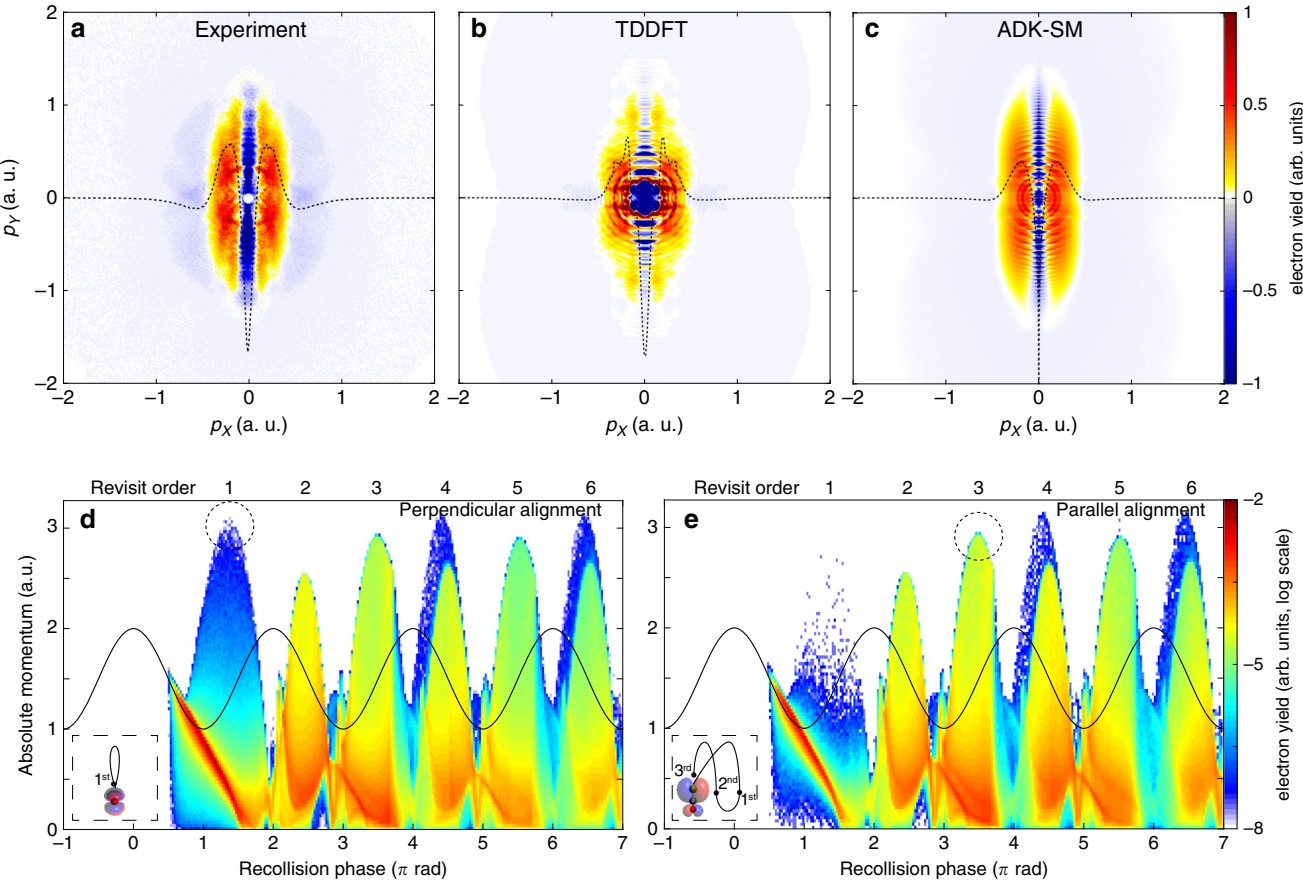

**Fig. 3 Differential momentum distributions and simulated final absolute photoelectron momentum. a–c** Differential momentum distributions (parallel–perpendicular) from (**a**) the experiment, and the (**b**) TDDFT and (**c**) ADK-SM calculations. To estimate the difference of the transverse momentum component the signal is integrated along the $Y$ axis, shown by the black dashed lines. **d, e** Final absolute photoelectron momentum as a function of the recollision phase (bottom) and revisit order (top) for **d** perpendicular and **e** parallel alignment, calculated with ADK-SM. The colour scale maps the electron counts at every momentum-phase point. The dashed black circles highlight the largest-momentum electrons at the most probable revisit order for the two alignments. A distance of $r < 5$ atomic units between electron and point charge is interpreted as a collision and only electrons with exactly one collision are shown. The solid black line depicts the external electric field. The insets give pictorial representations of molecular-frame electron trajectories, where the cardinals represent the revisit order.

molecule. The resulting momentum distributions are reported in Fig. 3d and e for perpendicular and parallel alignment, respectively. They consist of broad peaks appearing every half cycle of the electric field at phases close to $(k + 1/2)\pi$, $k = 1, 2, 3 \ldots$, for which the electron collides with the molecular cation when the laser field's vector potential is maximum. The first recollision event, i.e., the first peak in Fig. 3d at a phase of $3\pi/2$, allows the electron to reach the largest momentum as expected in the classical theory[29]. This is close to the maximum asymptotic kinetic energy, i.e., the 10 $U_p$ cut-off. The peaks appearing later correspond to electrons that have initially missed and then revisited the ion at later times. These subsequent rescattering events are expected to lead to lower photoelectron energies[29]. At the same time, these multiple revisits are possible only due to the Coulomb attraction of the ionised molecule[29]. Since the current understanding and analysis of strong-field self-diffraction experiments only consider the photoelectron recollision on the first revisit[7,12], the relevance of Coulomb attraction is usually neglected. However, our results demonstrate that it is a crucial ingredient to correctly understand molecular-frame electron rescattering. Note that a small yield at large momentum (>3 a.u.) is visible for both alignments (Fig. 3d, e) at the fourth and the sixth revisit. These revivals, caused by Coulomb focussing, vide supra[39], are expected to vanish for imperfect linear polarisation of

the laser field, as usually occurring in any experiment. This explains why ADK-SM for parallel alignment has another cut-off around 10.7 $U_p$, as well as the more pronounced cut-off around 9.3 $U_p$ in Fig. 2i. Due to the subtle conditions of Coulomb focussing, this effect will not be further considered in the discussion below; it does not contradict any of our general conclusions.

In this framework, the largest absolute momentum for perpendicular alignment comes from the first rescattering event at a phase around $3\pi/2$ (see Fig. 3d and its inset), which yields the largest momentum ~3.15 a.u. This momentum corresponds to an asymptotic kinetic energy ~10.5 $U_p$ and thus explains the experimental observation of the 10 $U_p$ cut-off for perpendicular alignment (see the red marker in Fig. 2c). This rescattering event is attenuated by the imprinting of the nodal plane perpendicular to the molecular axis[17], as otherwise this first peak would not only correspond to the largest photoelectron momentum, but also to the most probable recollision event. This attenuation for perpendicular alignment is responsible for the rescattering at the third revisit, i.e., at a phase around $7\pi/2$, to play a major role at lower energies and for the build-up of a secondary cut-off at ~9 $U_p$ (Fig. 2i). While this second distinct minimum is not clearly visible in the first derivative of the experiment and the TDDFT calculations, i.e., the red curves in Fig. 2c, f, the broad shape of the

minima in Fig. 2c, f at high energy may be in fact a signature of the attenuation of the scattering at first revisit and the relevant contribution of the third revisit. In the case of parallel alignment, instead, the first rescattering event is strongly suppressed and most of the large-momentum electrons come from the third revisit at a phase of $7\pi/2$, depicted in the inset of Fig. 3e; the fifth revisit also yields comparable momenta. As a result, the momentum cut-off is smaller, i.e., ~2.9 a.u. corresponding to a final kinetic energy of ~9 $U_p$. This is in good agreement with the experimentally observed reduced cut-off for parallel alignment (see the blue marker in Fig. 2c, f and i). This dynamics is mainly driven by the molecular potential: Here, the node of the HOMO along the laser polarisation imprints an angle on the electron emission at tunnelling. For OCS this angle was estimated to be ~30° with respect to the longitudinal $Y$ axis by the TDDFT calculations. This angle prevents the electron from rescattering at the first revisit. However, then the Coulomb attraction of the ionised molecule forces the electron to stay in the interaction region and to recollide at later revisits. It is important to note that the angle of emission and the rescattering at the $n$-th revisit are strongly correlated. Indeed, larger emission angles lead to later revisits and vice versa. As the consequence, the photoelectron cut-off carries a clear signature of the electronic structure at tunnelling. This angular dependence imprinted in the momentum distribution of the initial electron wavepacket leads to the breakdown[17] of the common product ansatz in QRS[15], where the initial and the rescattered parts of the wavepacket are separated and only the latter is considered angularly dependent in the recollision frame.

Furthermore, the photoelectron cut-off in the molecular frame carries crucial time information: While the cut-off for perpendicular alignment is strongly shaped by electrons recolliding 3/4 of an optical cycle after ionisation, as usually assumed, this is not true for parallel alignment: the cut-off is dominated by electrons revisiting the molecule much later, namely one or multiple optical cycles later. For a wavelength of 1300 nm, this corresponds, at least, to a delay of ~4.3 fs and it linearly increases with the wavelength. From Fig. 3d and e, apart from the aforementioned effects of Coulomb focussing, it is also evident that even-numbered revisits yield lower kinetic energies <8 $U_p$[29]. Since the time spent by the photoelectron before rescattering is usually exploited as the elementary delay step for time-resolved self-diffraction experiments[7], the use of this lower range of photoelectron energy[9,12] results in any time information being smeared out on much longer timescales. Furthermore, the analysis performed here demonstrated that this delay step strongly depends on the molecular-frame alignment and that the molecular potential sets a complex time–energy encoding in the electron dynamics. This molecular-frame clock for electron recollision could clearly be exploited to disentangle the structural dynamics with few-fs or even sub-fs temporal resolution. For instance, signals from the first (few) revisit order(s) could be selected in the experiment with near-single-cycle (few-cycle) laser pulses.

We demonstrated, experimentally and computationally, that the molecular frame determines the momentum distribution of high-energy rescattered electrons in strong-field ionisation. The basic concept of molecular-frame strong-field ionisation is captured by considering the initial conditions imposed by the molecular potential in the dynamics of the photoelectron. Furthermore, from the analysis of the rescattering trajectories it is evident that the molecular interaction plays a crucial role in setting a clock for the emission and the dynamics of high-energy electrons. It highlights that the molecular frame has a strong impact on the relation between the photoelectron energy and the rescattering time. This finding redefines the delay step of time-resolved self-diffraction experiments and opens up a perspective

on time-resolved diffraction experiments. These conclusions hold similarly for other observables related to electron recollision (e.g., high-harmonic-generation spectroscopy).

Our result represents an important benchmark for any self-diffraction measurement and represents a breakdown of the usual interpretation of LIED experiments[7,9,12]. We note that in such experiments mid-infrared lasers ($\lambda \approx 3\,\mu m$) are typically employed. Since the electron's excursion length increases with increasing laser wavelengths, we expect our findings to be even more relevant for actual LIED experiments.

Our study highlights the molecular-frame conditions as a crucial ingredient of self-diffraction experiments. This framework is general and can, in principle, be extended to any molecular system. Furthermore, the molecular-frame strong-field interaction was quantitatively modelled here by a fully interacting-electron TDDFT calculation and, in conjunction, by a semi-classical single-active-electron theory. We exploited TDDFT to evaluate the contribution of different molecular orbitals to the ionisation-rescattering dynamics as a benchmark for the applicability of the semi-classical approach. We expect this double-sided theoretical framework to become more and more important with increasing molecular complexity, where modelling the photoelectron dynamics may go beyond the capabilities of a single-orbital picture. In general, this also opens the perspective to investigate electron-correlation-driven phenomena in molecular strong-field physics[4]. Furthermore, the earliest moments of a strong-field interaction are intrinsically imprinted in the initial conditions of the photoelectron and in the final energy distribution. Thus, molecular-frame strong-field-ionisation experiments, in principle, allow one to achieve a deeper understanding of electron tunnelling, for instance, regarding the tunnelling time, and to track the molecular potential in real-time.

### Data availability
The data that support the findings of this study are available from the corresponding author upon reasonable request.

### Code availability
The OCTOPUS code is available from http://www.octopus-code.org. The code for the ADK-SM calculations is available from Jochen Küpper (jochen.kuepper@cfel.de) upon reasonable request.

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

## Acknowledgements

This work has been supported by the Clusters of Excellence "Center for Ultrafast Imaging" (CUI, EXC 1074, ID 194651731) and "Advanced Imaging of Matter" (AIM, EXC 2056, ID 390715994) of the Deutsche Forschungsgemeinschaft (DFG), by the European Research Council under the European Union's Seventh Framework Programme (FP7/2007–2013) through the Consolidator Grant COMOTION (ERC-Küpper-614507) and under the Horizon 2020 Research and Innovation Programme through the Advanced Grant QSpec-NewMat (ERC-Rubio-694097), and by the Helmholtz Association Initiative and Networking Fund. A.T. and J.O. gratefully acknowledge fellowships by the Alexander von Humboldt Foundation.

## Author contributions

A.T., S.T. and J.K. conceived the experiment. A.T., S.T., J.F.O., J.W., T.M. and J.O. performed the experiment. A.T. performed the data analysis. J.W. set up and performed the ADK-SM simulations. U.D.G. and B.F. performed the TDDFT calculations, and together with S.-K.S. and A.R. provided theoretical support. A.T., S.T., U.D.G., J.W., A.R. and J.K. interpreted the results and prepared the paper. All authors contributed to the discussion of the results and commented on the paper.

## Competing interests

The authors declare no competing interests.
