## [Peer Review File · Nature Communications]

Thanks for the detailed replies from the authors. I am satisfied with most of the responses and am very glad to recommend the manuscript to be published in *Nature Communication*. However, I would like the authors to address the following questions more clearly.

In my previous review report, my main concern was the “poor” agreement between the experimental and the ADK-SM simulated spectra. The authors argue that the main source of discrepancy is due to imperfect polarization and imperfect alignment of the molecular frame in the experiment. This is a reasonable argument, though I think the imperfect in laser polarization can be easily addressed by applying polarizer(s). And the authors account for imperfect molecular alignment in the TDDFT simulations but not in the ADK-SM simulations, why? Is it due to computational cost? For the case of perpendicular alignment, the authors propose that the broad shape of the first derivative minima at high energy may be a signature of the third revisit, but could it come from imperfect molecular alignment?

Regarding the impact of this study on time-resolved self-diffraction (e.g. LIED) experiments. The parameters of the lasers used for LIED are quite different from the one used in this work. For LIED, typically long wavelength lasers are used and very importantly, the re-collision energy of the electron is high (>100 eV). Can the authors comment on this question: Over what laser intensity and wavelength ranges should we concern that molecular alignment may affect the revisiting orders?

Reply to Reviewer

Thanks for the detailed replies from the authors. I am satisfied with most of the responses and am very glad to recommend the manuscript to be published in Nature Communication. However, I would like the authors to address the following questions more clearly.

We thank the Reviewer for the positive feedback on our work. In the next paragraphs we will address her/his questions.

In my previous review report, my main concern was the “poor” agreement between the experimental and the ADK-SM simulated spectra. The authors argue that the main source of discrepancy is due to imperfect polarization and imperfect alignment of the molecular frame in the experiment. This is a reasonable argument, though I think the imperfect in laser polarization can be easily addressed by applying polarizer(s).

We thank the Reviewer for this comment. We agree that a polarizer can be used to clean the laser polarization. However, the extinction contrast of available polarizers (Glan or Thin films) is limited, usually around 10^5 or 10^6 at best. Furthermore, the laser beam typically goes through optical pieces in transmission, such as vacuum windows, just before reaching the interaction point. The above-mentioned optics may introduce a small ellipticity, since they typically show a residual birefringence. SiO_2 , for example, has a $\Delta n \sim 0.01$ (at 600 nm). As a result, typical polarisation contrasts in strong-field experiments are around 10^3 or 10^4 , for which we can already expect strong-field features that deviate from the ones of a perfectly linearly polarised laser field.

Nevertheless, we agree that a more sophisticated experimental apparatus can be designed and realised to address the issue of experimental imperfect polarisation and improve the polarisation contrast, for example with in-vacuum polarisers. However, the above-mentioned setup would require a complete new design of the interaction region and the vacuum equipment and introduce severely increased further complexity in the experiment, which is beyond the scope of this work.

And the authors account for imperfect molecular alignment in the TDDFT simulations but not in the ADK-SM simulations, why? Is it due to computational cost?

We clearly acknowledge this important comment. In fact, the computational cost of the ADK-SM calculations is a lot lower than that of the TDDFT approach. The difficulty lies rather in the model itself. While TDDFT considers ‘all’ the relevant physics from the beginning, the ADK-SM model by design only contains the essential, manually added aspects of the strong-field ionisation process. By allowing imperfect alignment, the ionising laser-electric field would break the symmetry of the molecule, which would require an unproportional amount of extra theoretical effort to describe the initial ionisation step. We introduced the ADK-SM model description in order to obtain a mechanical picture of the electron’s continuum dynamics and to identify its crucial facets. That is why we refrained from going these (many) extra miles. The holistic computational description is already covered by the TDDFT setup.

For the case of perpendicular alignment, the authors propose that the broad shape of the first derivative minima at high energy may be a signature of the third revisit, but could it come from imperfect molecular alignment?

Imperfect alignment does not significantly affect the final radial momentum from a distinct revisit. However, it may slightly change the ratio between first and third revisit (and all the others). Therefore, imperfect alignment actually supports the interpretation that the broad shape of the first derivative minima for perpendicular alignment carry a signature of the third revisit.

Regarding the impact of this study on time-resolved self-diffraction (e.g. LIED) experiments. The parameters of the lasers used for LIED are quite different from the one used in this work. For LIED, typically long wavelength lasers are used and very importantly, the re-collision energy

of the electron is high (> 100 eV). Can the authors comment on this question: Over what laser intensity and wavelength ranges should we concern that molecular alignment may affect the revisiting orders?

As the electron's excursion length increases and the re-collision probability strongly decreases with larger laser wavelengths, we expect our findings to be even more relevant for actual LIED experiments in the mid-infrared regime.

We updated the manuscript to explicitly include this important point by adding the following sentence in the revised manuscript on line 440:

We note that in such experiments mid-infrared lasers ($\lambda \approx 3 \mu\text{m}$) are typically employed. Since the electron's excursion length increases with increasing laser wavelengths, we expect our findings to be even more relevant for actual LIED experiments. Our study [...]